# A pilot study of bladder voiding with real-time MRI and computational fluid dynamics

**Ryan Pewowaruk**[1], **David Rutkowski**[2,3], **Diego Hernando**[3,4], **Bunmi B. Kumapayi**[5], **Wade Bushman**[5], **Alejandro Roldán-Alzate**[1,3,6]*

**1** Biomedical Engineering, University of Wisconsin–Madison, Madison, WI, United States of America, **2** Cardiovascular Research Center, University of Wisconsin–Madison, Madison, WI, United States of America, **3** Radiology, University of Wisconsin–Madison, Madison, WI, United States of America, **4** Medical Physics, University of Wisconsin–Madison, Madison, WI, United States of America, **5** Urology, University of Wisconsin–Madison, Madison, WI, United States of America, **6** Mechanical Engineering, University of Wisconsin–Madison, Madison, WI, United States of America

* roldan@wisc.edu

**Data Availability Statement:** All relevant data are within the manuscript and its Supporting Information files.

## Abstract

Lower urinary track symptoms (LUTS) affect many older adults. Multi-channel urodynamic studies provide information about bladder pressure and urinary flow but offer little insight into changes in bladder anatomy and detrusor muscle function. Here we present a novel method for real time MRI during bladder voiding. This was performed in a small cohort of healthy men and men with benign prostatic hyperplasia and lower urinary tract symptoms (BPH/LUTS) to demonstrate proof of principle; The MRI urodynamic protocol was successfully implemented, and bladder wall displacement and urine flow dynamics were calculated. Displacement analysis on healthy controls showed the greatest bladder wall displacement in the dome of the bladder while men with BPH/LUTS exhibited decreased and asymmetric bladder wall motion. Computational fluid dynamics of voiding showed men with BPH/LUTS had larger recirculation regions in the bladder. This study demonstrates the feasibility of performing MRI voiding studies and their potential to provide new insight into lower urinary tract function in health and disease.

## Introduction

Lower urinary track symptoms (LUTS) and changes in bladder function occur frequently as individuals age [1–3]. Studies have evaluated the anatomical and functional changes of the bladder in patients with LUTS [4, 5]; however, the biomechanical characteristics of the lower urogenital tract, and how these are altered in patients with LUTS, are not fully understood. There is a compelling need to better delineate the anatomic and functional changes of the lower urinary tract in individual patients to improve diagnostic precision and allow for individualized treatment.

Lower urinary tract function is commonly assessed through multi-channel urodynamic studies that determine bladder pressure and flow during voiding. These studies can be performed in combination with fluoroscopic imaging to visualize the urine flow during voiding. However, these studies are invasive and provide little insight into the changes in bladder

**Funding:** This work was supported by the National Institutes for Health (nih.gov) grants K12DK100022 (AR, DH, WB) and T32 HL 007936 (RP, DR). The authors also wish to acknowledge support from GE Healthcare who provides research support to the University of Wisconsin. CONVERGE licenses were provided by a partnership between Convergent Science Inc. and the University of Wisconsin – Madison. The funders had no role in study design, data collection and analysis, decision to publish, or preparation of the manuscript.

**Competing interests:** The authors have read the journal's policy and have the following potential competing interests: the authors received CONVERGE licenses through a partnership between Convergent Science Inc. and the University of Wisconsin – Madison. This does not alter our adherence to PLOS ONE policies on sharing data and materials. There are no patents, products in development or marketed products associated with this research to declare.

anatomy and detrusor muscle function that occur with aging and lower urinary tract obstruction [6]. Use of non-invasive methods for the study of lower urinary tract anatomy and function has been limited. Image based patient specific computational models have been extensively used for cardiovascular evaluation and personalized treatment planning [7–10]. We envision a comparable approach for the evaluation of patients with LUTS. In this pilot study, we describe a magnetic resonance imaging (MRI) urodynamics method, as well as a patient specific MRI-based computational fluid dynamics (CFD) simulation of bladder voiding. These methods were applied to a small cohort of healthy men and men with benign prostatic hyperplasia and lower urinary tract symptoms (BPH/LUTS) to demonstrate proof of principle.

## Materials and methods

This study was approved by the University of Wisconsin Health Sciences IRB (approval number 2017-1373-CP006). Three men with BPH/LUTS were recruited from the University of Wisconsin Urology clinic (ages 73, 71, and 54). The inclusion criteria were adult men recently diagnosed with BPH. Three control subjects were recruited from a database of healthy controls maintained by the MR research group at UW-Madison (ages 66, 42, and 44). Inclusion criteria were healthy adult men not experiencing any symptoms consistent with BPH. Exclusion criteria for both groups were contraindication to MRI (e.g. pacemaker, contraindicated metallic implants, claustrophobia, etc) and patients who have undergone prostatectomy. MRI was performed on a clinical 3T scanner (Premier, GE Healthcare, Waukesha, WI) using a high-density flexible surface coil array (AIR Coil, GE Healthcare). An MRI urodynamics protocol was implemented which involved a fluid challenge and voiding during MRI. Three-dimensional 'Fast-spin echo' (FSE) T2-weighted acquisitions were performed immediately before and after voiding. A sagittal plane 2D spoiled gradient echo (SGRE) dynamic real-time imaging (RTI) acquisition was performed during voiding. The MRI protocol timeline is shown in Fig 1. The bladder was segmented from pre- and post- voiding 3D images of the bladder, while bladder cross-sectional area change during voiding was calculated from the 2D RTI. Measurements from both the 3D and 2D images were incorporated in a patient specific simulation of bladder voiding. Further details on the fluid challenge, 3D FSE MRI, 2D RTI MRI, post-processing and computational fluid dynamics are in the subsequent sub-sections.

### Fluid challenge

The subject was instructed to fast for 4 hrs. prior to arrival to the MRI session. Upon arrival, the subject was asked to empty his bladder and to drink approximately 1 L of a water and

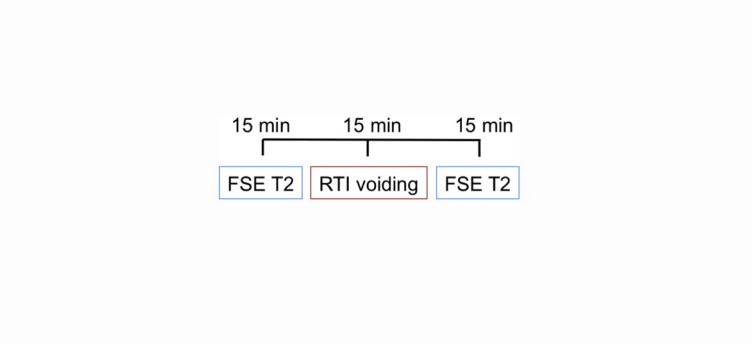

**Fig 1. MRI protocol.**

electrolyte beverage (Gatorade G2, The Gatorade Company, Chicago, IL) 30 minutes before MRI scanning, and an additional 100 ml every 10 minutes until scanning (total of approximately 1.3 L). Before the imaging session started, the subject was equipped with a condom catheter system that allowed him to void supine in the scanner. The condom catheter was connected to a container through a flexible plastic tube.

### 3D MRI of the lower urinary tract

2D fast-spin echo (FSE) T2-weighted axial acquisitions are currently used for clinical evaluation of patients with bladder or prostate-related diseases [11, 12]. Even though these images provide relevant information for clinical diagnosis, they are limited by their slice thickness (~7mm), demanding additional assumptions when performing volumetric measurements. Anatomical changes of the bladder can be spatially heterogeneous, requiring a volumetric (three-dimensional) evaluation. As an alternative, and in order to improve the spatial resolution for the current study, a single-slab 3D FSE T2-weighted imaging sequence was used [13]. This methodology utilizes parallel imaging–simultaneous acceleration in two directions–allowing the acquisition of additional and thinner slices, producing voxels with the same size in the slice and in-plane directions (isotropic resolution), contrary to conventional axial FSE acquisitions (anisotropic resolution, i.e.: slice thickness larger than in-plane pixel size).

### Dynamic imaging of the bladder

Advanced MRI sequence development has allowed for dynamic real-time acquisition of data for anatomical and functional assessment. Dynamic anatomical MRI has been used extensively for heart function assessment, where balanced steady-state-free precession (bSSFP) images have been shown to provide sufficient contrast to separate the myocardium and the blood pool [14]. bSSFP has previously been used to determine the bladder motion during voiding in healthy volunteers [15]. We endeavored to reproduce this protocol and obtain bSSFP images in a healthy volunteer during voiding. However, the volunteer expressed significant abdominal discomfort due to peripheral nerve stimulation during the real time MR acquisition. This made this approach untenable for use in our study. To circumvent this road block, we used an MRI sequence that consists of a series of parallel sagittal 2D spoiled gradient echo (SGRE) dynamic real-time images capturing the bladder neck and urethra as well as other regions of the bladder. In this protocol, images are constantly acquired for four minutes allowing complete capture of voiding mechanics during bladder voiding without inducing any discomfort. Dynamic images were segmented to measure the relative displacement of the bladder wall during voiding. Only one control subject underwent the real-time imaging protocol. The other two control subjects were imaged prior to the real-time imaging being added to the IRB protocol and instead were asked to step out of the scanner and void in the bathroom after the 3D MRI was performed. Immediately after voiding the subject was asked to return to the MRI scanner and the 3D MRI was repeated.

### MRI post-processing

Using 3D FSE T2-weighted images the bladder was segmented (Mimics, Materialise, Leuven, Belgium) for both pre and post voiding. The pre and post voiding bladder volumes were then exported as stereolithography (STL) files. From 2D RTI data during voiding, the area of a sagittal plane through the bladder was measured over time.

To estimate the motion of the bladder wall during voiding a spherical coordinate system was defined for the bladder, similar to previous motion estimation algorithms used in cardiac chambers [16, 17]. The coordinate system origin was set to be the center of the post voiding

bladder volume. In general, the bladder wall displacement (**d**) is a three dimensional vector that has spatial and time dependence.

$$[d_r, d_\theta, d_\phi] = f(\theta, \phi, t) \tag{1}$$

Based on prior work in cardiac chambers, we simplify the complete description of bladder wall motion by assuming the bladder wall only moves radially ($d_\theta = d_\phi = 0$) and the spatial and time dependence of the wall motion can be separated as

$$d_r = d_0(\theta, \phi)\alpha(t) \tag{2}$$

where $d_0(\theta,\phi)$ is the total displacement from the pre to post void anatomies and $\alpha(t)$ is the time dependence function that varies from 0 at the start of voiding to 1 at the end of voiding. For wall displacement analysis, the bladder wall was divided into anterior-posterior, dome-base, and left-right regions (Fig 2) and an asymmetry ratio was calculated based on the difference between the median displacement of the left and right bladder wall regions.

For each point on the bladder surface $d_0$ the distance between the pre and post voiding anatomies was calculated using a fast, minimum storage ray-triangle intersection algorithm [18] implemented in MATLAB. The time dependence function $\alpha(t)$ can be calculated from real time measurements of bladder area during voiding,

$$\alpha(t) = \left( \frac{A(t) - A(t_0)}{A(t_{end}) - A(t_0)} \right)^{\frac{1}{2}} \tag{3}$$

where $A(t)$ is the bladder area, $t_0$ is the time at the start of voiding and $t_{end}$ is the time at the end of voiding. Bladder area measurements from the real time sagittal MR images showed a sigmoidal behavior (Fig 3). Based on that behavior, $\alpha(t)$ was chosen to be a square root of cosine function.

## Computational fluid dynamics

The patient-specific bladder anatomies were imported into the computational fluid dynamics (CFD) software CONVERGE v2.4 (Convergent Science Inc, Madison, WI). Bladder wall motion was estimated as described above and imposed with a user-defined function to virtually drive voiding. As only one control subject had real-time MRI the same $\alpha(t)$ function was used for all three control subjects. The urethra wall was assumed to be rigid and the urethra outlet was set to atmospheric pressure. CFD simulations with large boundary motion are typically time-consuming, but this study employed two complementary strategies to enhance simulation speed. First, a cut-cell Cartesian mesh was used as these meshes have advantages in biomedical flows with moving boundaries [19]. Compared to typical boundary fitted meshes, cut-cell grids rapidly re-mesh to handle the moving bladder wall while simultaneously minimizing numerical diffusion from mesh motion. Second, the efficient re-meshing capabilities of cut-cell Cartesian meshes were used by adapting the mesh to the instantaneous flow field, which minimizes simulation time while maintaining accuracy. Results figures were generated with Tecplot 360 (Tecplot, Bellevue, WA). Vorticity, the curl of the velocity field, was averaged over the bladder volume (urethra not included). Dimensionless vorticity was then calculated based on average urethra flow rates and the prostatic urethra diameter for each subject.

## Results

The MRI urodynamics protocol was successfully completed in three healthy volunteers and three patients with BPH/LUTS. 2D planes of the bladder during voiding are shown in Fig 3,

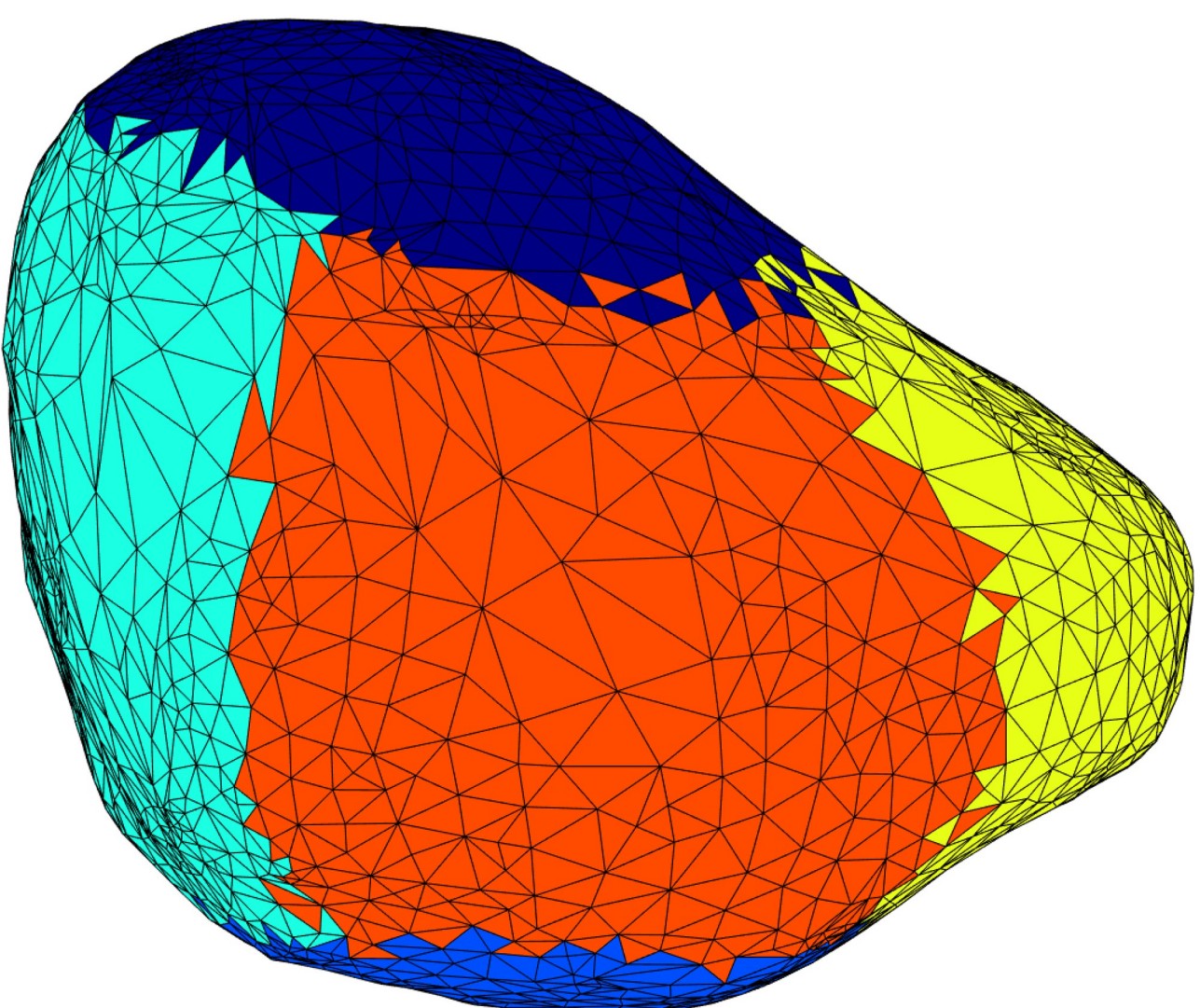

**Fig 2. Schematic of the bladder wall divided into anterior-posterior, dome-base, and left-right regions for regional displacement and asymmetry analysis.** Different colors represent different regions (anterior–aqua, posterior–yellow, dome–navy blue, base–royal blue, left–red, right–not shown).

along with the bladder cross-section area over time. The rate of area change is slowest at the beginning and end of voiding and greatest during the middle of voiding. Three-dimensional plots of the estimated bladder displacement throughout voiding at various time points are shown in Fig 3. The estimated displacement maps show that greatest displacement occurs at the dome of the bladder.

MRI urodynamics was performed on two additional healthy volunteers and three men with BPH/LUTS. The pre and post-voiding anatomies, estimated displacement maps and box plots showing the regional displacement behavior for each subject are shown in Fig 4. The control subjects had large displacements at the bladder dome with little observed asymmetry. The men with BPH/LUTS had smaller displacements and unlike the controls did not exhibit a consistent displacement pattern. These qualitative observations from the displacement maps were confirmed looking at probability functions of the displacement and the left-right asymmetry (Fig 5). Overall the bladder walls of men with BPH/LUTS moved only 25%-50% as much as the

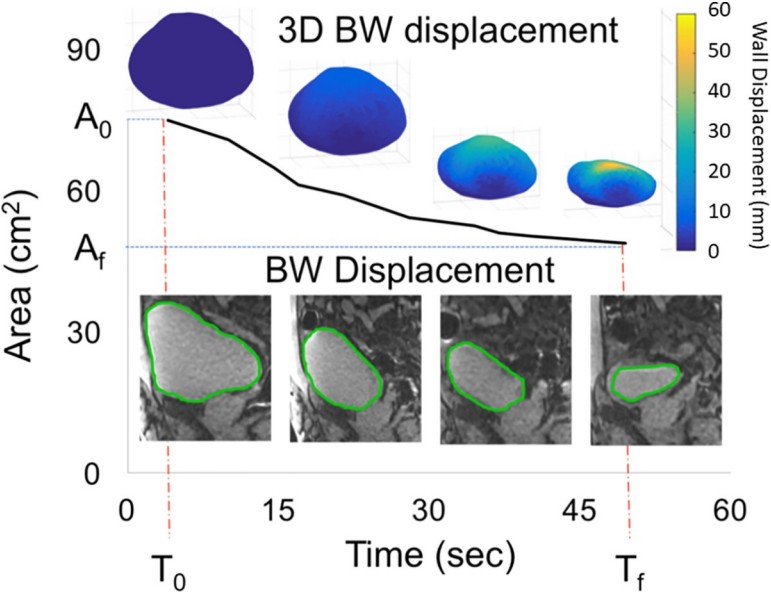

**Fig 3. Real time imaging of voiding in a 66-year-old healthy volunteer.** The curve shows the bladder emptying with respect to time following a sigmoidal behavior. 2D mid sagittal plane images show the bladder deformation at four different time points during the voiding event. Similarly, three-dimensional (3D) maps show bladder contraction (wall displacement) estimated from computational interpolation between pre and post void MRI at four different time points during voiding.

control subjects. The control subjects had little left-right asymmetry (4%-14%) while the BPH patients had large left-right asymmetry (40%-160%).

Urodynamics results from computational fluid dynamics (CFD) are shown in Fig 6 for a sagittal plane near the center of the bladder. Control subjects had higher urine velocities in the

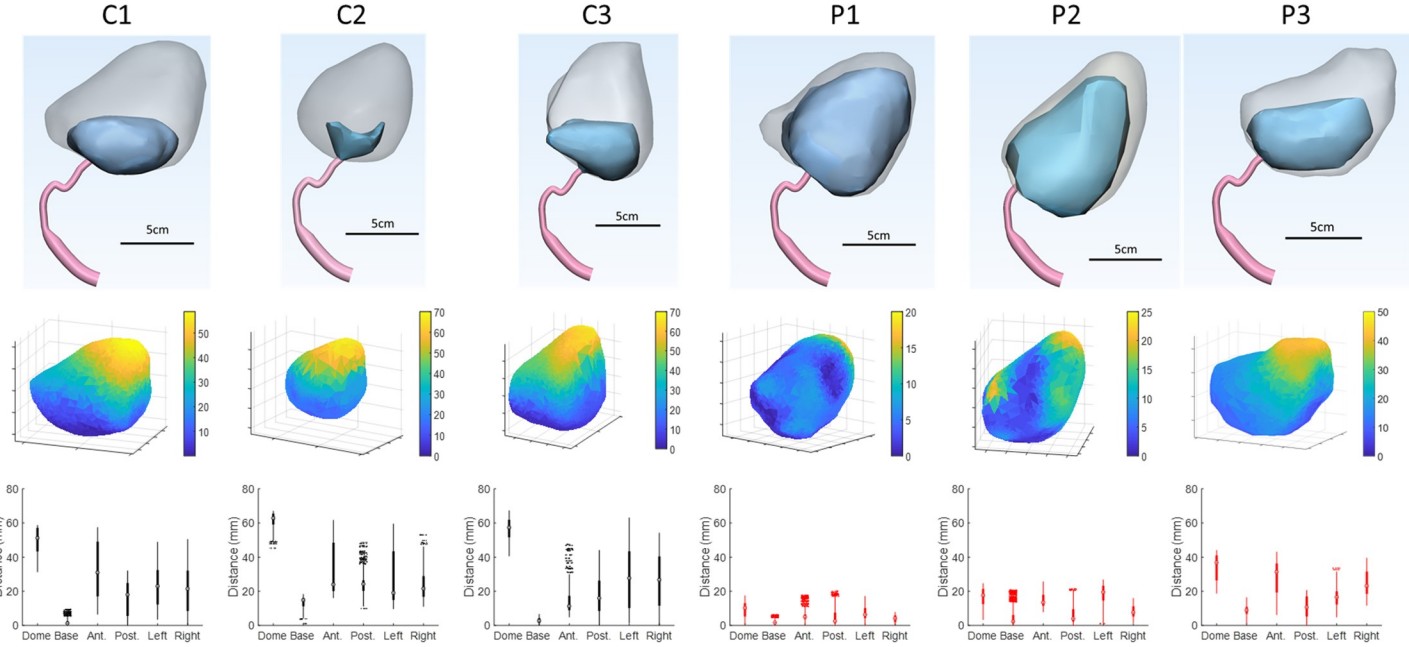

**Fig 4. Top row: Pre- and post-voiding bladder anatomies for each subject.** Middle row: Bladder wall displacement maps (in mm) for each subject. Note that the legend scale is much smaller for the men with BPH/LUTS. Bottom row: Box plots showing regional displacement behavior for each subject (C: Control; P: Patient).

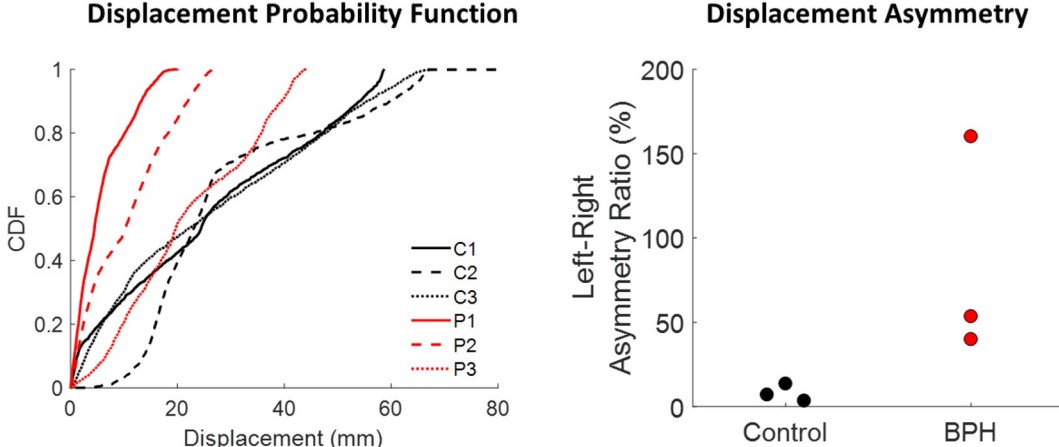

**Fig 5. Probability density functions of bladder wall displacement show the control subjects bladder move much more during voiding.** Additionally, the men with BPH/LUTS exhibit significant left-right asymmetry that was not observed in the control subjects.

bladder than men with BPH/LUTS due to their greater bladder wall displacements. Near the initiation of voiding, streamlines (showing the direction of urine flow) for all three control subjects and two men with BPH/LUTS (P1 and P3) were directed toward the bladder neck and prostatic urethra. Patient P2 has a small vortex just above the urethra. Towards the end of the voiding process, control subjects C1 and C3 had small recirculation regions by the anterior bladder wall while the streamlines for C2 were still all directed toward the bladder neck and prostatic urethra. Near the end of voiding patient P1 had a small posterior recirculation region, P2 had a large anterior recirculation region and P3 had large anterior and posterior recirculation regions. Despite the men with BPH/LUTS having larger recirculation regions, they had lower average vorticity (Fig 7) in the bladder due to their slower flow rates and smaller velocities. After making vorticity dimensionless (a normalization to account for differing flow rates between patients), vorticity was similar between healthy controls and men with BPH/LUTS.

## Discussion

This study demonstrates the feasibility of MRI bladder voiding studies to provide new insight into lower urinary tract function in health and disease, and to generate patient-specific simulations of voiding. Additionally, our studies have revealed a previously unsuspected aspect of bladder voiding. Is has been generally assumed that the sphere-shaped full bladder contracted with relatively uniform displacement. Yet, the results of this study suggest that this is not the case: Displacement analysis of voiding in healthy controls revealed a much greater displacement of the bladder dome than the other regions of the bladder (Fig 3). This is a simple, but profoundly startling, observation that may influence understanding of the contractile function of the bladder during voiding after further study. There is increasing appreciation for the role of impaired bladder contractility and its effects on treatment outcomes [20–24]. All of the men with BPH/LUTS in this study exhibited decreased bladder contractility on traditional multichannel urodynamic evaluation (data not shown). The limitation of standard urodynamic evaluation is that it provides only an indirect assessment of bladder contractility calculated from voiding pressure and flow at a single point in the voiding effort (maximum flow). MRI urodynamics, on the other hand, reveals significantly decreased and asymmetric bladder wall motion during voiding.

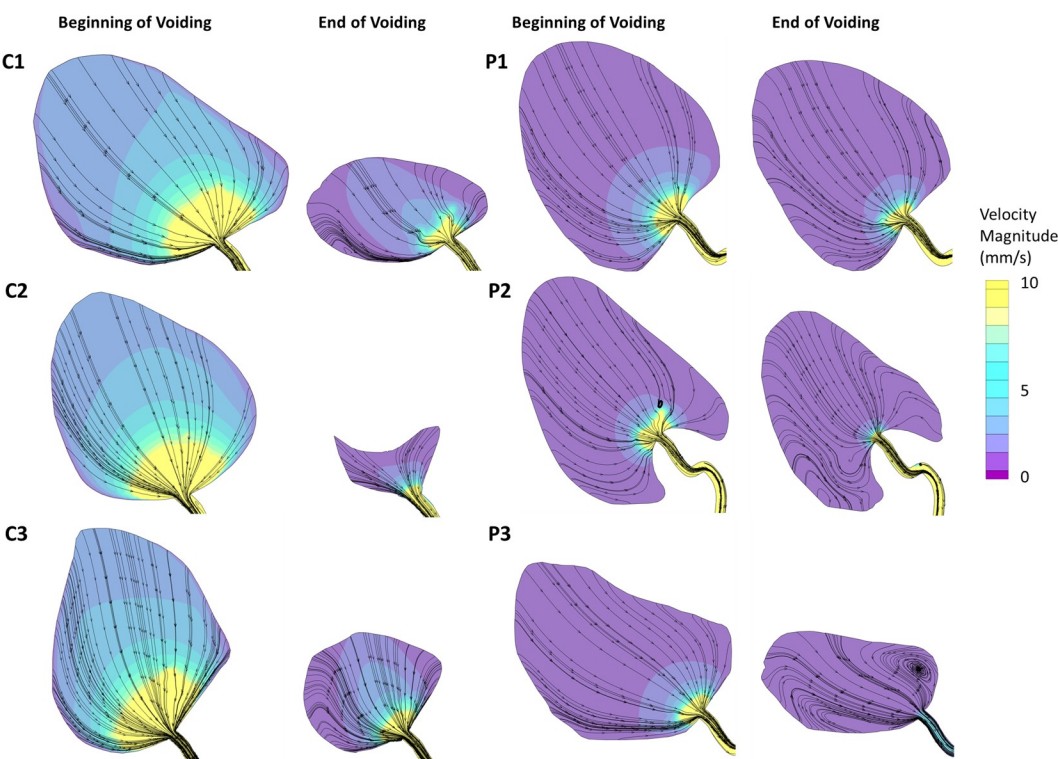

**Fig 6. CFD results showing velocity contours and streamlines on a sagittal plane at the center of the bladder for each subject.** Time points are displayed both near the initiation and termination voiding.

Although simulations of cardiac flow have incorporated wall motion going back to the 1970s [25, 26], this is to our knowledge the first MRI urodynamic-based simulation of bladder wall motion during voiding. Prior CFD studies have set the bladder wall to be an inlet [27, 28] or used the ureters to drive voiding [29]. The CFD methodology presented here represents a significant step towards improving the physical and physiological realism of urodynamic

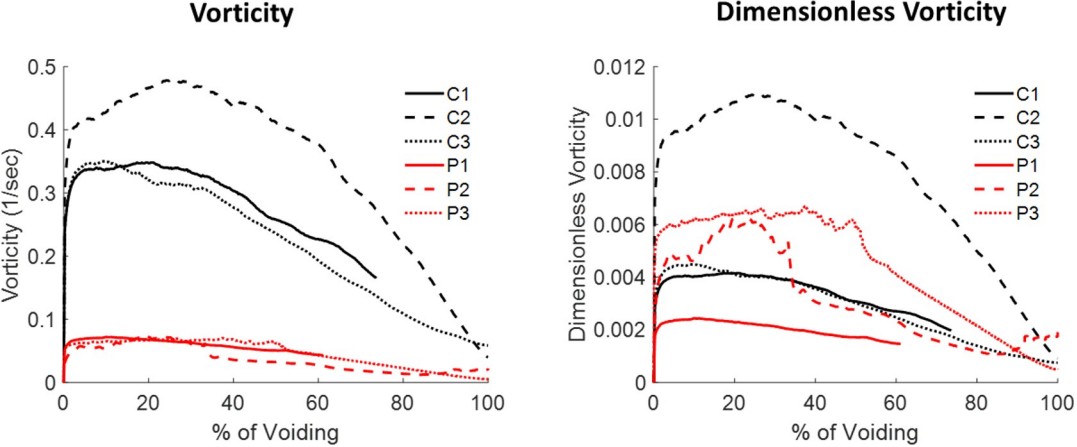

**Fig 7. Bladder vorticity was lower for the men with BPH/LUTS due to their lower flowrates and urine velocities.** After making vorticity dimensionless (a normalization to account for different flow rates between subjects), vorticity was similar for both groups.

simulations. As expected based on smaller bladder wall displacements, our simulation results showed that men with BPH/LTUS had lower urine velocities. More interestingly, the men with BPH/LUTS had larger recirculation regions in the bladder which could increase the energy demands of voiding.

In summary, this study demonstrated the feasibility of MRI bladder voiding studies to non-invasively investigate bladder function. Results from displacement analysis showed men with BPH/LUTS had decreased and asymmetric bladder wall motion compared to healthy male controls and fluid dynamic analysis of voiding showed them to have larger recirculation regions in the bladder.

## Supporting information

**S1 Data.**
(XLSX)

## Author Contributions

**Conceptualization:** Diego Hernando, Wade Bushman, Alejandro Roldán-Alzate.

**Data curation:** Ryan Pewowaruk, David Rutkowski.

**Funding acquisition:** Diego Hernando, Wade Bushman, Alejandro Roldán-Alzate.

**Investigation:** Bunmi B. Kumapayi.

**Methodology:** Ryan Pewowaruk, David Rutkowski, Diego Hernando, Bunmi B. Kumapayi, Alejandro Roldán-Alzate.

**Software:** David Rutkowski.

**Writing – original draft:** Ryan Pewowaruk.

**Writing – review & editing:** Ryan Pewowaruk, David Rutkowski, Diego Hernando, Wade Bushman, Alejandro Roldán-Alzate.

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
