## [Decision Letter · Decision Letter 0]

4 May 2020

PONE-D-20-10274

MRI Based Patient Specific Urinary Flow Dynamics Simulation

PLOS ONE

Dear Dr. Roldán-Alzate,

Thank you for submitting your manuscript to PLOS ONE. After careful consideration, we feel that it has merit but does not fully meet PLOS ONE’s publication criteria as it currently stands. Therefore, we invite you to submit a revised version of the manuscript that addresses the points raised during the review process.

We would appreciate receiving your revised manuscript by Jun 18 2020 11:59PM. To enhance the reproducibility of your results, we recommend that if applicable you deposit your laboratory protocols in protocols.io, where a protocol can be assigned its own identifier (DOI) such that it can be cited independently in the future. For instructions see: http://journals.plos.org/plosone/s/submission-guidelines#loc-laboratory-protocols

We look forward to receiving your revised manuscript.

Kind regards,

Robert Hurst, PhD

Academic Editor

PLOS ONE

Journal Requirements:

"The research presented was supported by the NIH (K12DK100022) and the UW CVRC T32 HL 007936

(RP and DR). The authors also wish to acknowledge support from GE Healthcare who provides research

support to the University of Wisconsin."

"National Institutes for Health (nih.gov)

K12DK100022 (AR, DH, WB)

T32 HL 007936 (RP, DR)

The funders had no role in study design, data collection and analysis, decision to

publish, or preparation of the manuscript."

Additionally, because some of your funding information pertains to commercial funding, we ask you to provide an updated Competing Interests statement, declaring all sources of commercial funding.

In your Competing Interests statement, please confirm that your commercial funding does not alter your adherence to PLOS ONE Editorial policies and criteria by including the following statement: "This does not alter our adherence to PLOS ONE policies on sharing data and materials.” as detailed online in our guide for authors  http://journals.plos.org/plosone/s/competing-interests.  If this statement is not true and your adherence to PLOS policies on sharing data and materials is altered, please explain how.

Please include the updated Competing Interests Statement and Funding Statement in your cover letter. We will change the online submission form on your behalf.

Additional Editor Comments (if provided):

As editor, I concur completely with the review as presented. At present it is a case report. In order for this to be useful to the field, additional data need to be obtained. I also feel comfortable with a single review, given that the paper is premature at this point. Some difficulties in obtaining reviewers was experienced due to the highly technical nature of the methodology. Most urologists are unfamiliar with the technical details of MRI, and I did not wish to cause further delays in returning the manuscript. I do not see an additional review changing my assessment of the manuscript.

Reviewers' comments:

Reviewer's Responses to Questions

**Comments to the Author**

1. Is the manuscript technically sound, and do the data support the conclusions?

Reviewer #1: Yes

2. Has the statistical analysis been performed appropriately and rigorously? 

Reviewer #1: No

3. Have the authors made all data underlying the findings in their manuscript fully available?

Reviewer #1: No

4. Is the manuscript presented in an intelligible fashion and written in standard English?

Reviewer #1: Yes

5. Review Comments to the Author

Reviewer #1: This is an interesting paper that has applied cardiac wall movement analysis using MRI to assess bladder voiding. The data presented is a bit premature, as the technique was only assessed on one healthy volunteer. There is no statistical analyses to support their hypothesis that most of the bladder wall movement occurs in the dome region. Although the application of the MRI methodology is new for bladder voiding, the approach to measure cardiac wall movement is well documented. I applaud the investigators for attempting to use the methodology in the bladder, which requires such approaches to non-invasively assess its function. Although, the approach is not entirely non-invasive, as the placement of the catheter and voiding approach described seems like it would be cumbersome for patients with bladder disorders.

At the very least, this method needs to be assessed in multiple individuals to support the outcome that there are regional differences in bladder wall movement during voiding. Individuals with bladder voiding issues should have also been included, which would have been more relevant to the urology community.

6. PLOS authors have the option to publish the peer review history of their article (what does this mean?). If published, this will include your full peer review and any attached files.

Reviewer #1: No

---

## [Author Response · Author response to Decision Letter 0]

12 Aug 2020

We thank the reviewer and editor for their consideration and feedback regarding our manuscript. Based on reviewer and editor comments we have expanded our manuscript from a case report of one healthy subject to a small cohort of three healthy subjects and three subjects with lower urinary tract symptoms. As the manuscript has been entirely rewritten changes to the manuscript are not marked. Point by point responses to reviewer comments follow.

1. Is the manuscript technically sound, and do the data support the conclusions?

Reviewer #1: Yes

Agree.

2. Has the statistical analysis been performed appropriately and rigorously?

Reviewer #1: No

Given the small sample sizes (n=3 and n=3 for both groups) we did not perform statistical analysis. The purpose of this study was to demonstrate a novel technique to assess bladder function so we feel this is appropriate. Additionally, we have added “pilot study” to the manuscript title per the editors advice and given our small sample size.

3. Have the authors made all data underlying the findings in their manuscript fully available?

Reviewer #1: No

 The data points used to create the figures are now included in a supplementary excel file.

4. Is the manuscript presented in an intelligible fashion and written in standard English?

Reviewer #1: Yes

 Agree.

5. Review Comments to the Author

Reviewer #1: This is an interesting paper that has applied cardiac wall movement analysis using MRI to assess bladder voiding. The data presented is a bit premature, as the technique was only assessed on one healthy volunteer. There is no statistical analyses to support their hypothesis that most of the bladder wall movement occurs in the dome region. Although the application of the MRI methodology is new for bladder voiding, the approach to measure cardiac wall movement is well documented. I applaud the investigators for attempting to use the methodology in the bladder, which requires such approaches to non-invasively assess its function. Although, the approach is not entirely non-invasive, as the placement of the catheter and voiding approach described seems like it would be cumbersome for patients with bladder disorders.

At the very least, this method needs to be assessed in multiple individuals to support the outcome that there are regional differences in bladder wall movement during voiding. Individuals with bladder voiding issues should have also been included, which would have been more relevant to the urology community.

We have expanded our manuscript from a case report of one healthy subject to a small cohort of three healthy subjects and three subjects with lower urinary tract symptoms. Given the small sample size we did not perform statistical analysis.

6. PLOS authors have the option to publish the peer review history of their article (what does this mean?). If published, this will include your full peer review and any attached files.

Do you want your identity to be public for this peer review? For information about this choice, including consent withdrawal, please see our Privacy Policy.

Reviewer #1: No

We do not wish to make the peer review history public.

---

## [Decision Letter · Decision Letter 1]

14 Oct 2020

PONE-D-20-10274R1

A Pilot Study of Bladder Voiding with Real-Time MRI and Computational Fluid Dynamics

PLOS ONE

Dear Dr. Roldán-Alzate,

Thank you for submitting your manuscript to PLOS ONE. After careful consideration, we feel that it has merit but does not fully meet PLOS ONE’s publication criteria as it currently stands. Therefore, we invite you to submit a revised version of the manuscript that addresses the points raised during the review process.

Please carefully address the issues raised by Reviewer 2. 

We look forward to receiving your revised manuscript.

Kind regards,

Robert Evan Hurst, PhD

Academic Editor

PLOS ONE

Reviewers' comments:

Reviewer's Responses to Questions

**Comments to the Author**

1. If the authors have adequately addressed your comments raised in a previous round of review and you feel that this manuscript is now acceptable for publication, you may indicate that here to bypass the “Comments to the Author” section, enter your conflict of interest statement in the “Confidential to Editor” section, and submit your "Accept" recommendation.

Reviewer #1: All comments have been addressed

Reviewer #2: (No Response)

2. Is the manuscript technically sound, and do the data support the conclusions?

Reviewer #1: Yes

Reviewer #2: Partly

3. Has the statistical analysis been performed appropriately and rigorously? 

Reviewer #1: N/A

Reviewer #2: N/A

4. Have the authors made all data underlying the findings in their manuscript fully available?

Reviewer #1: Yes

Reviewer #2: Yes

5. Is the manuscript presented in an intelligible fashion and written in standard English?

Reviewer #1: Yes

Reviewer #2: Yes

6. Review Comments to the Author

Reviewer #1: The authors have expanded the study, which has improved the overall quality of the study. Although it would have been ideal to include more patients and conduct statistical analyses, I think the study still warrants being considered for publication.

Reviewer #2: Methods:

1. Authors should include how they documented patients in control group were asymptomatic (e.g. International Prostate Symptom Score or AUA symptom score).

2. How were the 3 BPH/LUTS patients selected from the urology clinic? (Randomly?, IPSS?, Traditional Urodynamics?)

3. Patient selection did not specifically exclude those with neurologic diseases that are known to effect bladder function, such as, diabetes, parkinsonism, etc., even though the control group showed no symptoms. The group with BPH/LUTS could also have complicating neurologic disorders. Therefore, I suggest that the authors specifically state that there were no histories of such diseases.

4. It would have been nice to have Total Prostatic Volumes and Transition zone volumes on at least the BPH/LUTS group.

5. Examination with patients in the supine position in not very physiologic and represents a distinct limitation of the experiment. (Unlike with cardiac CFD.) The potential effect of positioning should be brought up in the DISCUSSION section.

6. The models rely on many assumptions, some of which the authors accurately report

Results:

1. P9,line1 Statement should insert the words “may be” before “due” or amend to say “there is an association between wall displacement and increased flow velocity in the bladder”, since they have not done a large cohort to provide proof.

2. Fig. 6 Caption The authors might have provided more information on how to interpret “streaming lines” and “time points.” Engineers may be familiar with these concepts, but urologists, radiologists imagers and other readers probably need assistance. Also, “velocity magnitude” should be defined here or in the text of the manuscript.

3. Fig. 7 Caption Insert “likely” before “due”.

Discussion:

1. P10,line 3 The authors are “profoundly startled” by dome movement, but I would expect that because the base of the inferior detrusor is limited in movement by the thicker and stiffer stiffer overlying trigone. This would be expected to be exaggerated by gravity if the subject voids in the usual upright posture. A few examples of this are listed below:

[Assessment of movements of the different anatomic portions of the bladder, implications for image-guided radiation therapy for bladder cancer]. [French]

Evaluation des mouvements des differentes portions anatomiques de la vessie, implications pour la radiotherapie guidee par l'image pour les cancers de vessie.

Pan Q; Thariat J; Bogalhas F; Lagrange JL.

Cancer Radiotherapie. 16(3):167-78, 2012 May.

[Journal Article]

UI: 22365260

Authors Full Name

Pan, Q; Thariat, J; Bogalhas, F; Lagrange, J-L.

Cite My Projects Annotate

AB PURPOSE: To assess interfraction and intrafraction bladder wall movements in the different anatomic portions of the bladder. PATIENTS AND METHODS: Six patients were treated for prostate cancer with conformal irradiation. Daily online cone beam computed tomography was performed for repositioning and an additional one was performed following irradiation once weekly. Four craniocaudal levels were defined to calculate movements amplitudes compared to the scanner tracking: level 1 at the bladder neck, level 2 at mid-height of the bladder, level 3 at mid-height of the dome, level 4 at the apex in a distended bladder. Bladder height was also measured. RESULTS: On 198 daily cone beam computed tomographies, radial bladder right/left/anterior/posterior wall displacements at level 2 were 0.08 +/- 0.24, 0.11 +/- 0.33, 0.16 +/- 0.45 and 0.14 +/- 0.50 cm and at level 3 0.07 +/- 0.78, 0.18 +/- 0.98, 0.43 +/- 0.94 and 0.04 +/- 1.02 cm. Dome and neck displacements were 0.08 +/- 1.41 cm and 0.08 +/- 0.64 cm. Seventeen cone beam computed tomographies were done following irradiation. Radial bladder right/left/anterior/halfway up the trine wall displacements at level 2 before and after irradiation were 0.02+/-0.18, 0.01+/-0.30, 0.09 +/- 0.32 and 0.22 +/- 0.42 cm and at level 3 0.27 +/- 0.60, 0.37 +/- 1.15, 0.18 +/- 0.87 and 0.54 +/- 1.68 cm. CONCLUSION: Significant bladder wall displacements were observed on the anterior wall and upper portion of the bladder. Isotropic margins may not be sufficient to account for inter- and intrafraction bladder wall displacements at the latter levels. Tailored bladder anatomy-based anisotropic margins may be necessary to optimally spare the small intestine and to guaranty proper tumour coverage in case of bladder cancer. For upper bladder tumours, margins of over 2 cm would be necessary, which make them less adequate for external beam irradiation.

Assessment of Bladder Motion for Clinical Radiotherapy Practice Using Cine–Magnetic Resonance Imaging

Catherine A McBain;Vincent S Khoo;David L Buckley;Jonathan S Sykes;Melanie M Green;Richard A Cowan;Charles E Hutchinson;Christopher J Moore;Patricia M Price

ISSN: 0360-3016; DOI: 10.1016/j.ijrobp.2008.11.040

International journal of radiation oncology, biology, physics. , 2009, Vol.75(3),

2. p10 line 7 “thatt” typo

3. p10,l para 7 The authors might consider that CFD may also show vesicoureteral reflux, although it is not demonstrated in this study cohort.

REFERENCES: There are several incomplete or mistaken references

#13 Busse R incomplete reference

#28 Turkiye “Klinikleri” J Med Sci

SUMMARY

Pilot feasibility studies require a lesser level of evidence than more determinative research. The authors conceive of an image-based non-invasive technique that may substitute for traditional multichannel urodynamics. I would offer that their studies may provide different information rather than fully replace pressure-flow studies. The results of this study can make no conclusions other than proof of concept which it adequately does. I would recommend the authors re-phrase the final paragraph to say that the findings “in men with BPH/LUTS “suggest asymmetric bladder wall motion compared to healthy men…….”

7. PLOS authors have the option to publish the peer review history of their article (what does this mean?). If published, this will include your full peer review and any attached files.

Reviewer #1: No

Reviewer #2: No

---

## [Author Response · Author response to Decision Letter 1]

16 Oct 2020

We thank the reviewers for their consideration of our manuscript and for their expert advice on how to best improve our manuscript. Changes to the manuscript are tracked and point-by-point responses to reviewer comments are below.

Methods: 

1. Authors should include how they documented patients in control group were asymptomatic (e.g. International Prostate Symptom Score or AUA symptom score).

Healthy controls were recruited from a database in the department of Radiology. At time of consenting patients were asked about LUT symptoms.

“At the time of consenting it was confirmed that patients were not experiencing LUT symptoms.”

2. How were the 3 BPH/LUTS patients selected from the urology clinic? (Randomly?, IPSS?, Traditional Urodynamics?) 

Patients with known BPH/LUTS and scheduled for surgical treatment were recruited for the study.

3. Patient selection did not specifically exclude those with neurologic diseases that are known to effect bladder function, such as, diabetes, parkinsonism, etc., even though the control group showed no symptoms. The group with BPH/LUTS could also have complicating neurologic disorders. Therefore, I suggest that the authors specifically state that there were no histories of such diseases.

None of the patients had a history of neurological disease

“Three men with known BPH/LUTS who were scheduled for surgical treatment were recruited from the University of Wisconsin Urology clinic (ages 73, 71, and 54), none of whom had a history of neurological disease. The inclusion criteria were adult men recently diagnosed with BPH.”

4. It would have been nice to have Total Prostatic Volumes and Transition zone volumes on at least the BPH/LUTS group.

Prostate volumes are now included in the results section.

“Prostate volumes segmented from 3D FSE MRI were larger for BPH/LUTS patients (45, 80, and 106 mL) compared to healthy volunteers (18, 37, and 42 mL).”

5. Examination with patients in the supine position in not very physiologic and represents a distinct limitation of the experiment. (Unlike with cardiac CFD.) The potential effect of positioning should be brought up in the DISCUSSION section.

The potential effects of positioning are now discussed In the manuscript.

“Men normally void either standing or sitting down, and it would be ideal to perform the dynamic studies in a standing position, however, that is not possible due to the patient’s position in the MRI scanner. While multichannel urodynamic evaluation is usually performed with men either standing or sitting, it is often performed in supine position for patients with neurologic disease such as spinal cord injury. Previous studies have demonstrated modest effects of study position on quantitative measures of voiding pressure and urine flow, and ; however, there is no evidence that the anatomy and contractile function of the bladder are significantly altered by position.”

6. The models rely on many assumptions, some of which the authors accurately report

Additional assumptions are now reported.

“The urethra wall was assumed to be rigid and the urethra outlet was set to atmospheric pressure. Additionally, the urine density and viscosity were assumed to be the same for all subjects.”

Results:

1. P9,line1 Statement should insert the words “may be” before “due” or amend to say “there is an association between wall displacement and increased flow velocity in the bladder”, since they have not done a large cohort to provide proof.

“Urodynamics results from computational fluid dynamics (CFD) are shown in Figure 6 for a sagittal plane near the center of the bladder. Control subjects had higher urine velocities in the bladder than men with BPH/LUTS that may be due to their greater bladder wall displacements.”

2. Fig. 6 Caption The authors might have provided more information on how to interpret “streaming lines” and “time points.” Engineers may be familiar with these concepts, but urologists, radiologists imagers and other readers probably need assistance. Also, “velocity magnitude” should be defined here or in the text of the manuscript.

“CFD results showing velocity contours and streamlines on a sagittal plane at the center of the bladder for each subject. Results are displayed for time frames from near both the initiation and termination voiding. Streamlines indicate the direction of urine flow and velocity magnitude is the magnitude of the velocity vector or the overall speed of urine flow.”

3. Fig. 7 Caption Insert “likely” before “due”.

“Bladder vorticity was lower for the men with BPH/LUTS, likely due to their lower flowrates and urine velocities. After making vorticity dimensionless (a normalization to account for different flow rates between subjects), vorticity was similar for both groups.”

Discussion:

1. P10,line 3 The authors are “profoundly startled” by dome movement, but I would expect that because the base of the inferior detrusor is limited in movement by the thicker and stiffer stiffer overlying trigone. This would be expected to be exaggerated by gravity if the subject voids in the usual upright posture. A few examples of this are listed below:

 [Assessment of movements of the different anatomic portions of the bladder, implications for image-guided radiation therapy for bladder cancer]. [French] 

Evaluation des mouvements des differentes portions anatomiques de la vessie, implications pour la radiotherapie guidee par l'image pour les cancers de vessie.

Pan Q; Thariat J; Bogalhas F; Lagrange JL. 

Cancer Radiotherapie. 16(3):167-78, 2012 May.

[Journal Article]

UI: 22365260 

Authors Full Name

Pan, Q; Thariat, J; Bogalhas, F; Lagrange, J-L.

 Cite My Projects Annotate

AB PURPOSE: To assess interfraction and intrafraction bladder wall movements in the different anatomic portions of the bladder. PATIENTS AND METHODS: Six patients were treated for prostate cancer with conformal irradiation. Daily online cone beam computed tomography was performed for repositioning and an additional one was performed following irradiation once weekly. Four craniocaudal levels were defined to calculate movements amplitudes compared to the scanner tracking: level 1 at the bladder neck, level 2 at mid-height of the bladder, level 3 at mid-height of the dome, level 4 at the apex in a distended bladder. Bladder height was also measured. RESULTS: On 198 daily cone beam computed tomographies, radial bladder right/left/anterior/posterior wall displacements at level 2 were 0.08 +/- 0.24, 0.11 +/- 0.33, 0.16 +/- 0.45 and 0.14 +/- 0.50 cm and at level 3 0.07 +/- 0.78, 0.18 +/- 0.98, 0.43 +/- 0.94 and 0.04 +/- 1.02 cm. Dome and neck displacements were 0.08 +/- 1.41 cm and 0.08 +/- 0.64 cm. Seventeen cone beam computed tomographies were done following irradiation. Radial bladder right/left/anterior/halfway up the trine wall displacements at level 2 before and after irradiation were 0.02+/-0.18, 0.01+/-0.30, 0.09 +/- 0.32 and 0.22 +/- 0.42 cm and at level 3 0.27 +/- 0.60, 0.37 +/- 1.15, 0.18 +/- 0.87 and 0.54 +/- 1.68 cm. CONCLUSION: Significant bladder wall displacements were observed on the anterior wall and upper portion of the bladder. Isotropic margins may not be sufficient to account for inter- and intrafraction bladder wall displacements at the latter levels. Tailored bladder anatomy-based anisotropic margins may be necessary to optimally spare the small intestine and to guaranty proper tumour coverage in case of bladder cancer. For upper bladder tumours, margins of over 2 cm would be necessary, which make them less adequate for external beam irradiation.

Assessment of Bladder Motion for Clinical Radiotherapy Practice Using Cine–Magnetic Resonance Imaging

Catherine A McBain;Vincent S Khoo;David L Buckley;Jonathan S Sykes;Melanie M Green;Richard A Cowan;Charles E Hutchinson;Christopher J Moore;Patricia M Price

ISSN: 0360-3016; DOI: 10.1016/j.ijrobp.2008.11.040

International journal of radiation oncology, biology, physics. , 2009, Vol.75(3), 

This statement has been removed and we these studies are now included in our discussion section.

“Results from daily CTs of bladder cancer patients over the course of a week (presumably at varied levels of bladder filling) similarly show that the bladder wall displacements were greater for the dome of the bladder than the base [20]”

“Asymmetric bladder wall motion has previously been identified in bladder cancer patients during bladder filling [26].”

2. p10 line 7 “thatt” typo

This typo has been corrected.

3. p10,l para 7 The authors might consider that CFD may also show vesicoureteral reflux, although it is not demonstrated in this study cohort.

Investigation of vesicoureteral reflux would require our models to include the ureters. This is now discussed in the manuscript.

“Further improvements to the simulation realism by including the ureters could expand CFD analysis to include phenomena such as vesicoureteral reflux.”

REFERENCES: There are several incomplete or mistaken references

#13 Busse R incomplete reference

#28 Turkiye “Klinikleri” J Med Sci

Reference #13 has been completed. Reference #28 (now #30) has been double checked and it is correct. 

SUMMARY

Pilot feasibility studies require a lesser level of evidence than more determinative research. The authors conceive of an image-based non-invasive technique that may substitute for traditional multichannel urodynamics. I would offer that their studies may provide different information rather than fully replace pressure-flow studies. The results of this study can make no conclusions other than proof of concept which it adequately does. I would recommend the authors re-phrase the final paragraph to say that the findings “in men with BPH/LUTS “suggest asymmetric bladder wall motion compared to healthy men…….”

The summary has been rewritten

“In summary, this pilot study demonstrated the feasibility of MRI bladder voiding studies to non-invasively investigate bladder function. Results from displacement analysis suggested that men with BPH/LUTS have decreased and asymmetric bladder wall motion compared to healthy male controls and fluid dynamic analysis of voiding suggested that men with BPH/LUTS have larger recirculation regions in the bladder.”

---

## [Editor Report · Decision Letter 2]

19 Oct 2020

A Pilot Study of Bladder Voiding with Real-Time MRI and Computational Fluid Dynamics

PONE-D-20-10274R2

Dear Dr. Roldán-Alzate,

We’re pleased to inform you that your manuscript has been judged scientifically suitable for publication and will be formally accepted for publication once it meets all outstanding technical requirements.

Kind regards,

Robert Evan Hurst, PhD

Academic Editor

PLOS ONE
---

## [Editor Report · Acceptance letter]

3 Sep 2020

PONE-D-20-10274R1 

A Pilot Study of Bladder Voiding with Real-Time MRI and Computational Fluid Dynamics 

Dear Dr. Roldán-Alzate:

I'm pleased to inform you that your manuscript has been deemed suitable for publication in PLOS ONE. Congratulations! Your manuscript is now with our production department. 

Kind regards, 

on behalf of

Dr. Robert Evan Hurst 

Academic Editor

PLOS ONE